# A mixed-method exploration into the experience of members of the FAO/WHO International Food Safety Authorities Network (INFOSAN): study protocol

Carmen Joseph Savelli,[1,2] Céu Mateus[2]

[1]Food Safety and Zoonoses, World Health Organization, Geneva, Switzerland
[2]Division of Health Research, Faculty of Health and Medicine, Lancaster University, Lancaster, UK

**Correspondence to**
Mr. Carmen Joseph Savelli;
savellic@who.int,
c.savelli@lancaster.ac.uk

## ABSTRACT

**Introduction** The International Food Safety Authorities Network (INFOSAN) is a global network of national food safety authorities from 188 countries, managed jointly by the Food and Agriculture Organization of the United Nations (FAO) and the World Health Organization (WHO), which facilitates the rapid exchange of information during food safety related events. The proposed research will interrogate INFOSAN in order to describe and explore the experiences of members and better understand the role of the network in mitigating the burden of foodborne illness around the world.

**Methods** Examined through a community of practice lens, a three-phase research design will combine quantitative and qualitative methods (including website analytics in phase 1, online survey administration in phase 2 and semistructured interviews in phase 3) to elicit a broad and deep understanding of the network operation and member experiences.

**Analysis** In phases 1 and 2, quantitative data collected from the INFOSAN Community website and the online questionnaires will be analysed using descriptive summary statistics. In phase 3, interpretative phenomenological analysis will be used to engage in a dialogue with study participants to explore and describe their lived experiences regarding participation in activities related to INFOSAN. An important aspect of the overall analysis will be triangulation of the information collected from each phase, including quantitative indicators and qualitative value stories, in order to provide a robust understanding of member experience.

**Ethics and dissemination** This study has undergone ethical review and has received approval from Lancaster University's Faculty of Health and Medicine Research Ethics Committee, as well as the ethics review committee of the WHO. Findings from the study will be disseminated as a PhD thesis submitted to Lancaster University. In addition, results of the research shall be submitted for publication to relevant academic or professional conferences and journals or other media, including books or websites.

## Strengths and limitations of this study

► This study represents the first ever to explore and describe the experiences of all International Food Safety Authorities Network members with respect to their participation in network activities as a means to improve global food safety and to prevent foodborne illness, using a range of data collection methods.
► The mixed-method approach will strengthen the credibility of the findings and provide a more complete view and deeper understanding of the experiences of members.
► The main limitation of this study is that the online survey will be available in English, French and Spanish only, and interviews will be conducted in English only due to limited resources of the study team.

first time. Together, 31 foodborne hazards are estimated to cause 600 million cases of foodborne disease and 420 000 deaths annually, worldwide.[1] Foodborne diseases are preventable, but ensuring a safe national food supply requires a robust food control system and coordination among different government sectors responsible for human health, animal health, agriculture, trade and others. In addition, as a global commodity, food produced in one country can readily cause international outbreaks if contaminated food is exported abroad. Channels of communication on matters of food safety must therefore be well established within and between countries in order to prevent national and international food safety emergencies.[2] It is for these reasons why the WHO launched the International Food Safety Authorities Network (INFOSAN) in 2004. Today, the overall aim of INFOSAN is to halt the international spread of contaminated food, prevent foodborne disease outbreaks and strengthen food safety

## INTRODUCTION

In 2015, the WHO reported estimates of the global burden of foodborne diseases for the

systems globally.[3] The four main objectives of INFOSAN are to (1) promote the rapid exchange of information during food safety-related events; (2) share information on important food safety related issues of global interest; (3) promote partnerships and collaboration between countries and between networks and (4) help countries strengthen their capacity to manage food safety emergencies. Using the INFOSAN Community Website (ICW; a secure, online portal), INFOSAN members from around the world exchange information on urgent food safety issues and emerging trends of potential global interest. The ICW also provides a virtual environment to share lessons learned and allows members to pose questions to one another for the purpose of building knowledge related to food safety.[3]

INFOSAN members have a common identity that is defined by a shared domain of interest. By joining the network, each has committed to taking actions that contribute to a safer global food supply by engaging in joint activities and discussions to facilitate knowledge transfer and exchange (KTE). Common responsibilities are also shared by members, as defined by the INFOSAN Secretariat. Combined, these common responsibilities and activities create a sense of community and are undertaken with the intention of facilitating the application of best practices to improve food safety. In addition, INFOSAN members are each practitioners in their respective countries, some as food regulators, risk analysts, epidemiologists or other professionals. While their focus may be different, the uniting factor is that their practice, in some respect, aims to make contributions towards the reduction of foodborne illness. It is the shared domain, community and practice that allow for INFOSAN to be understood as a community of practice (CoP).[4] A CoP is a group of people sharing a particular concern, problem or passion for an area and who deepen their knowledge and expertise by learning from one another and interacting on a regular basis.[4] Such interactions may occur in person or through technology-mediated means, as with INFOSAN.

A growing body of research suggests that KTE can be effectively fostered within CoPs, leading to the uptake and application of best practices by individuals and teams in various sectors, including health, business and beyond.[5] Multiple systematic reviews[6–9] suggest that fostering a virtual or electronic CoP among professionals in public health is useful for encouraging KTE, which translates into adoption of evidence-based best practices and, by extension, improved public health. Rajić et al[10] have described the benefits of facilitating KTE among food safety professionals working at the intersection of agriculture and health. Together, the literature suggests that a CoP like INFOSAN, connecting food safety and public health professionals from around the world, is an appropriate tool to facilitate KTE in this area. However, while INFOSAN has been operating for nearly 15 years to facilitate the aforementioned activities among its members, it has never been characterised or examined as a functional CoP, and its value, as understood from the perspective of its members, has never been determined in a systematic or rigorous way. Furthermore, a paucity of research has been conducted to investigate the attributes and effectiveness of specific tools or CoPs like INFOSAN to facilitate cross-border communication during international food safety events. To date, most of the publications mentioning such tools focus on summarising a particular incident response, rather than explicitly examining the tools that were used. However, such reports of international food safety events commonly conclude with recommendations to better use existing international networks and communication tools to improve and expedite information exchange.[11–17] In addition, several published studies have specified the important role that INFOSAN has played in facilitating rapid international communication between government officials that has led to the timely implementation of risk management measures during a food safety emergency.[18–20]

## RESEARCH AIM
The overall aim of this study is to explore and describe the experiences of INFOSAN members with respect to their participation in network activities as a means to improve global food safety and prevent foodborne illness.

## RESEARCH OBJECTIVES
1. Assess the functioning of INFOSAN as a CoP by obtaining systematic insights into the characteristics, performance and opinions of members.
2. Gain a broad and deep understanding of members' perceptions of the use of INFOSAN as a global communication tool for KTE and the prevention of foodborne illness in their respective country.
3. Determine if participation in INFOSAN creates value for members and explore the mechanisms through which this may occur.

## MAIN RESEARCH QUESTIONS
1. How is the ICW being used to support the network activities?
2. What are the barriers and enablers to active participation in INFOSAN?
3. Do members of INFOSAN believe that participation in the network has prevented (or will prevent) foodborne illness in their country?
4. Does participation in INFOSAN create value for members and if so, through what mechanisms does this occur?

## METHODOLOGY AND METHODS
### Philosophical underpinnings
The theoretical framework for exploring experiences in CoPs proposed by Wenger, Trayner and de Laat[21] is used for this study and has its roots in critical realism. Critical realism is a philosophical perspective that accepts the

existence of stable and enduring features of reality independently of one's ability to perceive them and therefore should be measured as a sum of different perspectives.[22] This concept represents the ontological perspective of critical realism (where ontology should be understood as the philosophical study of being). Furthermore, critical realism asserts that social phenomena and their meanings are continually being created by social actors (ie, the philosophical concept of constructivism) and may differ depending on one's perspective (ie, the philosophical concept of relativism). Constructivism and relativism represent the epistemic orientation of critical realism (where epistemology is the philosophical study of knowledge and how one knows).[23]

The importance of using quantitative indicators, together with qualitative narratives from different perspectives in a community, is a concept described by Trayner and Wenger[21] in order to understand what kind of experiences create value within CoPs. Quantitative indicators can be readily obtained from most virtual CoPs by measuring the number of unique users logging in to a web portal over time or the number of postings within a discussion forum, for example. However, Wenger, Trayner and de Laat[21] suggest that relying on such indicators alone would require too many assumptions to be made to accurately find meaning in experiences and determine value. Such indicators should rather serve as a point of reference from which value-creation stories can be elicited from members directly in order to provide a more robust understanding of experience. Likewise, examining only qualitative narratives from members would ignore the opportunity to crosscheck information with indicators to determine how perceptions and actions correspond. It is this acknowledgement of multiple perspectives of reality that demonstrates congruence with critical realism. In taking this approach, discrepancies and correspondences can be observed, and both grounded narratives and aspirational narratives become illuminated (where grounded narratives are those for which value-creation stories are supported by quantitative indicators and aspirational narratives are not).[21]

With an understanding of INFOSAN as a functional CoP, and with an appreciation for the critical realist perspective that underpins its examination, it is incumbent on the researcher of the proposed study to take a mixed-method approach in order to strengthen the credibility of the findings and to provide a more complete view and deeper understanding of the experiences of members. The use of such an approach for the investigation of experiences of online community members is strongly supported by a growing body of literature.[24–26]

## Setting

The study will be conducted by the researcher from within the Department of Food Safety and Zoonoses (FOS) at the headquarters of the WHO in Geneva, Switzerland (where the INFOSAN Secretariat is based); however, the true setting is global since INFOSAN membership spans 188 countries.

## General comments on sampling

For each phase of the research, participants will include registered INFOSAN members only. These individuals have been officially designated by their national government and are all registered on the ICW (n=500). INFOSAN membership includes both men and women in approximately equal proportions.

## General comments on recruitment

Written permission to conduct the proposed research on INFOSAN has been received from the director of FOS at the WHO, where the INFOSAN Secretariat is based. To announce the launch of the study, all members of INFOSAN will receive by email a written introductory information about the proposed research (information email 1; online supplementary file 1), including an invitation to attend an online seminar (ie, webinar), delivered by the researcher, to find out more information about the study and ask any questions or seek clarification. This webinar, like all future webinars discussed in this proposal, will be scheduled at three different times to allow INFOSAN members across different time zones to participate. The webinar will be delivered using the secure online conferencing tool, WebEx. The live webinars will not be recorded, but a recording will be made by the researcher alone and will be made available to INFOSAN members to view in case they would like to do so at their convenience.

### Phase 1: analysis of ICW website access and usage
#### Sampling, recruitment and consent

Information email 1 will explain the three different phases of the proposed research and remind INFOSAN members that data analysed in phase 1 of the study will be extracted from the ICW in accordance with the terms and conditions of use that each member consented to when they registered online. The relevant text within the terms and conditions of use reads as follows: 'Utilization of website analytics tools or other methods will be applied periodically to summarise members' access to and usage of the ICW for monitoring, evaluation and research purposes.' As such, data pertaining to all members of INFOSAN, with respect to their access to and usage of the ICW, will be used in phase 1 of the study unless members expressly indicate their wish to be excluded. Information email 1 (and related webinars) will ensure that INFOSAN members understand that any members not wishing to have their website access and usage data used for the purpose of this study will have 2 weeks to make this indication by email to the researcher. All those members who do not otherwise object will be recruited for phase 1. Those who are recruited in phase 1 will have a further 2 weeks to opt out of the study, after which time this will no longer be possible due to aggregation of the data.

#### Data collection and analysis

Access to the ICW is granted to the researcher in his capacity as a staff member at WHO, and approval for use

in this research has been granted by the director of FOS, WHO. Data from the ICW will be collected retrospectively for the period between February 2012 (when the website was launched) and December 2018. Information from all recruited members concerning the following variables will be downloaded from the website, anonymised and exported into Microsoft Excel and SPSS v23 for analysis: type of member, sex, country, languages spoken, government sector, primary function (ie, risk assessment, risk communication or risk management) and areas of scientific expertise. These data have all been automatically collected and stored in the internal website database at the time of member registration. Additional data about length of membership, last access to the website, and discussion thread initiations, responses and views will also be collected and exported for analysis. Once collected, all anonymised data will be analysed using descriptive summary statistics, which will allow for stratification by a number of variables, including type of member, geographical region and length of membership. Analysing these data will provide an objective, foundational layer of information about the experiences of all members and will be triangulated with data from phases 2 and 3 to determine if members' reported attitudes and experiences reflect their online behaviours. In addition, triangulated data will also allow for INFOSAN to be described with respect to its stage of community development according to Wenger et al,[4] taking into account its structuring characteristics as described by Dube et al.[27]

### Phase 2: administration of an online survey
#### Sampling, recruitment and consent
On conclusion of data collection and analysis in phase 1, all members of INFOSAN will receive information email 2 (online supplementary file 2), which will include indicative results from phase 1 and an invitation to attend a webinar, delivered by the researcher, to learn more about the results from phase 1 and to provide further details about phase 2. Immediately following the webinar, all INFOSAN members (n=500) will receive information email 3 (online supplementary file 3), including the invitation to participate in phase 2 of the study and containing a link to an online survey. Only those who express consent will be recruited as participants for phase 2. No consent form will be collected because that would compromise anonymity; however, the email will explain that by clicking on the link, the participants confirm that they have read the introductory information and understand what is expected of them as participants in this phase of the study (full details in online supplementary file 3).

#### Data collection and analysis
Recruited participants will be requested to complete an online questionnaire that will take between 30 and 45 min. Responses to the questionnaire are intended to provide systematic insights into the characteristics, performance and opinions of INFOSAN members and

contribute to a broader understanding of their experiences. The questionnaire will consist of questions from the Community Assessment Toolkit (CAT),[28] as well as a supplemental set of questions, tailored specifically to INFOSAN members. During the development of the CAT, Verburg and Andriessen[28] demonstrated that the methodology for its development was based on dominant theories of CoPs and group dynamics. The CAT was pilot tested and studied with seven CoPs (n=271) to enable reliability tests and scale analysis. Cronbach's alpha for the scales ranged from 0.59 to 0.91, and as the items referred to quite separate goals, Verburg and Andriessen[28] reported these values acceptable. Using the CAT in this study will enable future comparative research between CoPs that have been assessed with the same tool (eg, Roberts[25]). However, given the unique nature of INFOSAN and the specific objectives of this study, it is also necessary to develop a short set of supplemental questions to examine the experiences of INFOSAN members that are unique to this particular CoP. A preliminary set of supplemental questions has been inserted to the appropriate sections of the CAT questionnaire. These supplemental questions will be reviewed for content validity by a panel of six experts consisting of members of the INFOSAN Advisory Group since they are familiar with the constructs that the supplemental questions are designed to measure. The expert panel will judge whether the supplemental questions adequately measure the construct they are intended to assess, and whether these supplemental questions are indeed sufficient to measure the domain of interest. A Content Validity Index (CVI) will be computed for each supplementary item, and items with a CVI of 0.78 or higher will be considered evidence of good content validity.[29] Based on feedback from the expert panel, the supplemental questions will be revised prior to pilot testing. After incorporating any suggestions from INFOSAN Advisory Group members, FAO/WHO Regional Food Safety Advisors/Officers will be requested to pilot test the supplemental questions. As a measure of reliability, internal consistency will be estimated using the coefficient alpha, also known as Cronbach's alpha. Determining the internal consistency will reflect the extent to which the questionnaire items are intercorrelated, or whether they are consistent in measurement of the same construct.[30]

Because the questionnaire (including the CAT questions and the supplementary questions) will be disseminated to INFOSAN members in 188 countries, it will be adapted from English into French and Spanish in order to encourage a higher response rate. The aim of the adaptation process is to achieve different language versions of the English instrument that are conceptually equivalent in both French and Spanish. The instrument should be equally natural and acceptable and should practically perform in the same way, with a focus on cross-cultural and conceptual equivalence, rather than on linguistic/literal equivalence. A well-established method to achieve this goal is to use forward translations and back translations.[31]

This method has been refined over the course of several WHO studies to result in the following process, which will be undertaken to adapt the questionnaires into French and Spanish.[31] First, one native Spanish speaker and one native French speaker from the Department of FOS at WHO will conduct forward translations of the entire questionnaire. These translators will be health professionals, familiar with the terminology of the area covered by the instrument. Next, two bilingual expert panels will be convened to identify and resolve the inadequate expressions/concepts of the translation, as well as any discrepancies between the forward translation and the English versions of the questions. Next, using the same approach as outlined in the first step, the questionnaire will then be translated back to English by an independent translator (from central translation services at WHO), whose mother tongue is English and who has no knowledge of the questionnaire. As in the forward translation, emphasis in the back translation should be on conceptual and cultural equivalence and not linguistic equivalence. Discrepancies should be discussed with the researcher and further work (forward translations and discussion by the bilingual expert panel) will be iterated as many times as needed until a satisfactory version is reached. This version will then be pre-tested with a group of 20 interns at WHO including 10 native French speakers and 10 native Spanish speakers. A call for volunteers will be disseminated through the WHO intern mailing list. Pretest respondents will be administered the questionnaire and then systematically debriefed. This debriefing will ask respondents what they thought the questions were asking, whether they could repeat the questions in their own words, and what came to their minds when they heard a particular phrase or term. The debriefing will take the form of a focus group, organised at the WHO. The adaptation process followed will be traceable through a number of documents, including (1) initial forward versions, (2) a summary of recommendations by the expert panels, (3) the back translations, (4) a summary of problems found during the pretesting of the instrument and the modifications proposed, (5) the final version and (6) a description of the samples used in this process (ie, the composition of the expert panel and the pretest respondent samples). The process of expanding and adapting the questionnaire is summarised in online supplementary file 4.

Quantitative data collected from the online questionnaires will be analysed using descriptive summary statistics, allowing for stratification by a number of variables, including type of member, geographical region and length of membership. A variety of techniques for univariate, bivariate and multivariate analyses using SPSS will be employed in order to examine patterns and relationships between variables. Depending on the response rate to the survey, the researcher may need to adjust for non-response bias in order to generalise the results to the entire network.[32] The researcher is a native English speaker with a working knowledge of French and basic knowledge of Spanish. As such, for the very few instances where an open text response is an option, answers provided in French or Spanish will be translated by the researcher with the aid of Google translate, if necessary.

## Phase 3: Semistructured interviews
### Sampling, recruitment and consent
On conclusion of data collection and analysis in phase 2, all members of INFOSAN will receive Information email 4 (online supplementary file 5), which will include indicative results from phase 2 and an invitation to attend a webinar, delivered by the researcher, to learn more about the results from phase 2 and to provide further details about phase 3. The email and webinar will also include a request that members interested in participating in phase 3 indicate this by emailing the researcher. The online survey in phase 2 will also serve as a recruitment tool for phase 3 since a concluding statement on the survey will indicate that members interested in participating in phase 3 of the research should contact the researcher separately by email.

A minimum of 6 and a maximum of 12 participants will be included in the study sample, aiming for two participants from each of the six geographical regions delineated by the WHO. The purpose is not to be representative of the entire network; however, because INFOSAN is global, including participants from each region may reveal a richer pool of experience than if all members were selected from a single region. In addition, a sample size within this range should allow for the examination of similarities and differences between individuals, without producing an overwhelmingly large amount of qualitative data that cannot be managed within the confines of the study timeline. This sample of participants will be restricted to those INFOSAN members who have been registered members for a minimum of 2 years at the time of their interview to ensure they have a reasonable level of experience with the network from which to draw. The sample will be limited to those members who speak English due to limited funding for research conduct (including for translation and interpretation) and limited time for collecting and analysing data in other languages. In the event that at least one member per WHO region does not volunteer after receiving information email 4, follow-up emails will be sent to those members from the regions where volunteers are still needed, to indicate as such.

If more than two people from a single WHO region indicate their interest in participating in phase 3, the two people who have been members the longest will be selected and notified by email from the researcher, providing they are not from the same country. Any individuals who volunteer for phase 3 but who are not selected will be emailed individually by the researcher and provided with an explanation about how the selection was made (online supplementary file 6).

Prior to commencing their interview, INFOSAN members who volunteer and are recruited for phase 3

of the study will need to have read, signed and returned by email to the researcher the phase 3 consent form as detailed in online supplementary file 7. Immediately before the interview, the researcher will ensure the participants are aware they can choose to withdraw from the study at any time, for any reason, before or during their interview and withdraw their data up to 2 weeks after their interview.

### Data collection and analysis

Recruited participants will be requested to participate in a semistructured interview conducted online using the secure tool, WebEx, because participants are anticipated to be located in various countries around the world (no face-to-face interviews will be conducted). The interviews will be scheduled to last between 45 min and 1 hour. The same four themes explored in phase 2 will be explored during the interviews, and the discussion will focus on answering the related questions of 'how?' and 'why?' instead of just 'what?' as elaborated in the interview guide in online supplementary file 8 (results from phase 1 and/or 2 may lead to the addition or deletion of certain questions from the current interview guide). The researcher will follow interpretative phenomenological analysis (IPA, a qualitative research approach that is used to explore and examine personal lived experiences[33]) to engage in a dialogue with study participants to explore and describe their lived experiences regarding participation in activities related to INFOSAN. Such a method requires a flexible data collection instrument,[33] and therefore, the semistructured format of the interviews will be conversational in style, allowing the researcher and the participant to engage in a dialogue where questions can be modified depending on responses. This format also enables the researcher to prompt further elaboration in certain areas of interest identified by participants, allowing for more flexibility than a structured interview.[32] The interviews will be audio-recorded and then transcribed by the researcher. Audio recordings will be made using WebEx and immediately downloaded, password protected and encrypted on a laptop. Audio-recorded data will be anonymised as far as is possible (given the nature of audio data) by saving the file with an de-identified tag (eg, participant 1, region X). Prior to recording, the researcher will remind the participants that they can refrain from using names of people and places to the extent possible when answering questions, to assist with anonymisation. Once recording has started, the researcher will not use the participant's name during the interview. Transcripts will be anonymised by replacing identifying names of people or places with a de-identified tag (eg, participant 1, region X). Analysis will involve multiple readings of the interview transcripts, coupled with note taking, followed by transformation of notes into emergent themes and, finally, seeking relationships and clustering themes.[32]

An important aspect of the overall analysis will be triangulation of the information collected from each phase, including quantitative indicators and qualitative value

stories. Anonymised information and quotations from participant interviews will be reported, representing the limits to confidentiality. The combined strengths of quantitative and qualitative methods can contribute to improved study validity, credibility and overall integrity, and provide a broad and deep understanding of members' experiences.[32 34 35] The indicative results from phase 3, along with a triangulated analysis of data from all three phases, will be communicated to INFOSAN members through information email 5 (Supplementary file 9) and a final webinar, delivered by the researcher.

## PATIENT AND PUBLIC INVOLVEMENT

Patients were not involved in the design of this study. The first author has had conversations with several INFOSAN members in finalising this study design. These conversations have followed a presentation of the research design at a regional meeting of INFOSAN members in Miami, USA, in November 2017, where INFOSAN members from approximately 30 countries were present. In addition, a more detailed presentation of the study design was delivered at a meeting of eight INFOSAN members in Geneva, Switzerland, in December 2017, where further discussion has contributed to the finalisation of the study design, including the overall aim, objectives and research questions. The questionnaire will also be vetted by several target participants (selected because they are also part of the INFOSAN Advisory Group) as described in online supplementary file 4. The results of each phase of the study will be communicated to study participants via email and through webinars delivered by the first author.

## ETHICAL ISSUES
### Data management and storage

In accordance with the General Data Protection Regulation (GDPR) and the (UK) Data Protection Act 2018, all electronic data will be anonymised and stored onto encrypted password-protected storage devices, including data keys and a laptop computer) provided by the WHO. These encrypted files will use meaningful file names and version numbers and will be accompanied by a Readme file. Personal identifiers will be kept separately from anonymised data in either encrypted computer storage (for electronic files) or in a locked filing cabinet (for paper files) and destroyed following the completion and acceptance of the PhD thesis. Only the researcher and his supervisors will have access to the anonymised data for the duration of the study. Electronic copies of transcripts will be transferred in person by the researcher from WHO headquarters to Lancaster University on encrypted, password-protected storage devices and then archived for 10 years in secure encrypted storage on the Lancaster University server, after which time they will be destroyed. Storage and destruction of data will be done by Lancaster University's designated data protection officer. The researcher may need to modify the aforementioned

arrangements related to data management and storage and privacy protection measures to ensure that provisions of the GDPR are accommodated in case such provisions are supplemented, interpreted or clarified while the study is being conducted.

## Conducting insider research

By the nature of this work-based research project, the researcher is an insider, investigating an issue that examines, in broad terms, the operation of an organisational programme. The researcher is therefore an agent of his organisation as a technical officer at WHO and also an agent of Lancaster University as a PhD student. As such, the ethical considerations for the design of this research project have been carefully made from the insider researcher perspective.

In addition to approval being granted by the director of FOS to conduct this research, the proposed study has been subject to scrutiny and approval by the Faculty of Health and Medicine Research Ethics Committee at Lancaster University and the WHO Research Ethics Board before it could commence. This process involved technical review by an external scientific committee of experts. In addition, the conduct of the research will be governed by the WHO Code of Ethics and Professional Conduct and the WHO Code of Conduct for Responsible Research, both of which emphasise the need for all research to be conducted with integrity, accountability, independence, impartiality, respect and professional commitment.

While there are several positive aspects to being an insider researcher (eg, informed perspective and ability to implement study recommendations directly), the potential conflicts of interest must be carefully considered, acknowledged and addressed. As members of INFOSAN are familiar with the researcher in his role at WHO, he will need to ensure transparency and clarity that the proposed research is being conducted as part of a PhD study. It will also be necessary to ensure that, despite having access to additional data or information, only those data collected with the expressed consent of participants are used and reported on for the purposes of this study. INFOSAN members must be assured that neither their participation nor abstention will impact their future treatment as an INFOSAN member or the technical support provided to them or their agency by the WHO.

Acknowledging the role of the researcher as an insider is congruent with the methodology used to conduct this research on INFOSAN through a CoP lens. IPA as a research methodology is concerned with carefully detailing the lived experience of individuals. As explained by Guldberg and Mackness,[26] using IPA to understand experience aligns with a CoP lens since the focus of analysis is on the interpretations of members and the values they attribute to them. In addition, IPA acknowledges and embraces the role of the researcher's interpretation and understanding of members' lived experiences. The researcher, through his insider role on the INFOSAN Secretariat, plays an important part in connecting, communicating and facilitating interaction among members, and therefore his familiarity with the participants and ability to provide expertise in the interpretation and understanding of members' experiences should be considered an asset.

While the role of insider researcher can present some potential conflicts of interest, transparency in process, due permissions from senior WHO staff and assurances given to INFOSAN members should ensure this research is conducted to the highest ethical standard. In addition, several techniques well known to insider researchers will be employed to aid in accurately understanding and documenting the experiences of INFOSAN members, including the practice of reflexivity. The practice of reflexivity will involve active engagement of the self and questioning of the researcher's own perceptions in order to expose their contextualised nature.[36] A diary will be kept by the researcher during data collection and analysis to document reflexivity including predictions of outcomes for each phase of the study and how they relate to one another. The diary will not include any identifying information related to participants to ensure anonymity of participants, and any reflections on interviewed participants will reference a de-identified tag (eg, participant 1, region X). The diary will be stored in a locked cabinet in the main researcher's office when not being used in case it contains any sensitive information.

## DISSEMINATION PLANS

Results from this study can be used directly by the INFOSAN Secretariat at WHO in order to plan future interventions intended to encourage and support active participation in this important global network. In addition, results from this study may also contribute new knowledge regarding the coordination of international virtual networks and communities of practice in the realm of global health. The researcher will therefore disseminate findings from the study in the following ways: (1) via webinars, open to all INFOSAN members following each of the three phases of the study; (2) as an oral presentation to staff within FOS, WHO and (3) as a PhD thesis submitted to Lancaster University. In addition, results of the research shall be submitted for publication to relevant academic or professional conferences and journals or other media including books or websites.

**Contributors** CJS conceived the original idea, designed the study, drafted the manuscript and approved the final document. CM drafted the manuscript and approved the final document. CJS is a staff member of the WHO. The authors alone are responsible for the views expressed in this publication and do not necessarily represent the views, decisions or policies of the WHO.

**Funding** This study is funded by the WHO.

**Competing interests** None declared.

**Patient consent for publication** Not required.

**Provenance and peer review** Not commissioned; externally peer reviewed.

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
