## [Reviewer comments · BMJ Open]

ARTICLE DETAILS

TITLE (PROVISIONAL)	A mixed-methods exploration into the experience of members of the FAO/WHO International Food Safety Authorities Network (INFOSAN): Study Protocol
AUTHORS	Savelli, Carmen; Mateus, Ceu

VERSION 1 - REVIEW

REVIEWER	Dr. Emma Griffiths Simon Fraser University 8888 University Drive, Burnaby, BC Canada
REVIEW RETURNED	06-Nov-2018

GENERAL COMMENTS	The manuscript entitled "A mixed-methods exploration into the experience of members of the FAO/WHO International Food Safety Authorities Network (INFOSAN): Study Protocol", describes a protocol for exploring the experiences of the participants of the INFOSAN foodborne health event information network using three different methods. The paper describing the protocol is thorough, providing an overview of the ethics, review process, data collection tools, proposed comparisons, theoretical background and purpose. The manuscript is well-written and will provide novel insight into the perceived efficacy of the network and its tools, community functioning, and initiatives. As such I recommend it for publication, with a few suggestions. 1. Most of the text is very practical, however there are a few parts that are more philosophical that are written in language that is harder to understand for public health and food research scientists (presumably the intended audience). Particularly, the discussion of critical realism on p7 (lines 3-9), and the discussion of IPA and how it relates to critical realism on p17 (31-49), are somewhat difficult to decipher. Some simplification would improve the readability of the text.2. Can the author address the issue of how privacy will be protected in the contextual diary that will be kept describing reflexive insights from the interviews? Much care has been taken to describe the steps taken to protect participant anonymity, as well as the resulting data from ICW usage and the questionnaire, but there was little describing steps to ensure that the information in the diary would be protected, which could possibly contain sensitive information.3. Can the author describe the steps that would be taken should the WHO, or other authoritarian body being consulted, decide the results should not be released (either due to perceived faults in the study, its interpretation, or other reasons)? Or if such bodies ask for changes which which the author disagrees - what will you do?
--

	4. Can the author describe whether they plan to make recommendations to INFOSAN and/or surveillance/response networks in general, based upon the results?
--	---

REVIEWER	Ching-Fu Lin National Tsing Hua University, Taiwan
REVIEW RETURNED	14-Jan-2019

GENERAL COMMENTS	The submitted study protocol aims to explore the experience of members of the FAO/WHO International Food Safety Authorities Network (INFOSAN) by applying a well elaborated and structured three-phase research design that includes quantitative and qualitative data analytics, Interpretative Phenomenological Analysis (IPA), among others. In addition to the research design of this study protocol, its relevant key literature, research objectives and questions, and methodology and methods have been adequately described in a balanced and complete manner. The only concern I have regarding the study protocol is the discussion on ethical issues (namely, data management and storage, conducting insider research, and dissemination plans). The study protocol seems to regard such ethical issues as static rather than dynamic issues. However, they are constantly changing against the backdrop of (uncertain and evolving) legal and institutional context. For example, the General Data Protection Regulation (GDPR) entered into effect last year, but many key provisions have yet to be supplemented, interpreted, and clarified - and may in turn affect the data management and storage and privacy protection measures proposed by the study protocol. Furthermore, the potential conflicts of interest problem of the researcher as an insider at the WHO INFOSAN, while currently adequately safeguarded, may get worse if some future incidents generate unnecessary pressure on the research or reputational risks to the WHO INFOSAN. Negative events as such might undermine the credibility and integrity of this study. Therefore, the research as an insider should, as appropriate, be institutionally separately from WHO INFOSAN so as to ensure a higher level of independence. An alternative may be to apply a stricter measure to actively and dynamically ensure the researcher's work as an insider in the WHO INFOSAN is not and will not be influenced by factors related to her or his research process and (potential) outcomes.
---

VERSION 1 – AUTHOR RESPONSE

Reviewer 1

Comment	Response
Most of the text is very practical, however there are a few parts that are more philosophical that are written in language that is harder to understand for public health and food research scientists (presumably the intended audience). Particularly, the discussion of critical realism on p7 (lines 3-9), and the discussion of IPA and how it relates to critical realism on p17 (31-49), are somewhat difficult to decipher. Some	Thank you for the suggestions. Regarding the discussion on critical realism (p7): I have simplified this section by including some explanations of the most unfamiliar words used to discuss the philosophical underpinnings of the study including ontology, epistemology, constructivism and relativism. In doing so, I believe that those who are unfamiliar with such

simplification would improve the readability of the text.	terms should be able to grasp the concepts and hopefully learn something new. Regarding the discussion on IPA (p17), I have removed the reference to critical realism in this section to simplify the paragraph since the focus is rather about how utilizing IPA to understand experience aligns with a CoP lens since the focus of analysis is on the interpretations of members and the values they attribute to them.
2. Can the author address the issue of how privacy will be protected in the contextual diary that will be kept describing reflexive insights from the interviews? Much care has been taken to describe the steps taken to protect participant anonymity, as well as the resulting data from ICW usage and the questionnaire, but there was little describing steps to ensure that the information in the diary would be protected, which could possibly contain sensitive information.	The following text has been added to explain how privacy will be protected re: the diary: “The diary will not include any identifying information related to participants to ensure anonymity of participants and any reflections on interviewed participants will reference a de-identified tag (e.g. Participant #1 Region X). The diary will be stored in a locked cabinet in the main researcher’s office when not being used in case it contains any sensitive information.”
3. Can the author describe the steps that would be taken should the WHO, or other authoritarian body being consulted, decide the results should not be released (either due to perceived faults in the study, its interpretation, or other reasons)? Or if such bodies ask for changes which the author disagrees - what will you do?	The study methodology has been approved by the director of the Department of Food Safety and Zoonoses, WHO as well as a panel of three external experts whose job was to review the technical robustness of the study methodology and identify any faults. If any faults are identified in the future that have not been previously considered or if challenges are made to the interpretation of collected data, the first author will consult with his PhD supervisor at Lancaster University and with the three external experts to discuss the legitimacy of such claims. Depending on the outcome of these discussions, adjustments may be made as necessary. However, if WHO asks for changes undermining the author’s research independence, these changes will not be made. This is unlikely to happen as WHO actually stipulates that all external publications authored by WHO staff include the following text: “The authors alone are responsible for the views expressed in this article and they do not necessarily represent the decisions, policy or views of the World Health Organization.” That text should have been included in the original submission and its omission was an oversight and has since been added.

4. Can the author describe whether they plan to make recommendations to INFOSAN and/or surveillance/response networks in general, based upon the results?	The results may inform both INFOSAN and other networks. I have added the following sentences in the “dissemination plans” section to make this explicit: “Results from this study can be utilized directly by the INFOSAN Secretariat at WHO in order to plan future interventions intended to encourage and support active participation in this important global network. In addition, results from this study may also contribute new knowledge regarding the coordination of international virtual networks and communities of practice in the realm of global health.”
--	---

Reviewer 2

Comment	Response
The only concern I have regarding the study protocol is the discussion on ethical issues (namely, data management and storage, conducting insider research, and dissemination plans). The study protocol seems to regard such ethical issues as static rather than dynamic issues. However, they are constantly changing against the backdrop of (uncertain and evolving) legal and institutional context. For example, the General Data Protection Regulation (GDPR) entered into effect last year, but many key provisions have yet to be supplemented, interpreted, and clarified - and may in turn affect the data management and storage and privacy protection measures proposed by the study protocol. Furthermore, the potential conflicts of interest problem of the researcher as an insider at the WHO INFOSAN, while currently adequately safeguarded, may get worse if some future incidents generate unnecessary pressure on the research or reputational risks to the WHO INFOSAN. Negative events as such might undermine the credibility and integrity of this study. Therefore, the research as an insider should, as appropriate, be institutionally separated from WHO INFOSAN so as to ensure a higher level of independence. An alternative may be to apply a stricter measure to actively and dynamically ensure the researcher's work as an insider in the WHO INFOSAN is not and will not be influenced by factors related to her or his research process and (potential) outcomes.	Thank you for raising these issues. Concerning the GDPR, I have added the following text to acknowledge the dynamic nature of the situation: “The researcher may need to modify the aforementioned arrangements related to data management and storage and privacy protection measures to ensure that provisions of the GDPR are accommodated in case such provisions are supplemented, interpreted or clarified while the study is being conducted.” Regarding potential conflicts of interest as an insider researcher from WHO, this is now addressed by the inclusion of the following disclaimer text: “The authors alone are responsible for the views expressed in this publication and they do not necessarily represent the views, decisions or policies of the World Health Organization.” That text should have been included in the original submission and its omission was an oversight and has since been added.

VERSION 2 – REVIEW

REVIEWER	Emma Griffiths University of British Columbia, Canada
REVIEW RETURNED	09-Mar-2019

GENERAL COMMENTS	This is an excellent protocol paper. It is written very clearly, with well articulated objectives and methods. My suggestions are very minor. Could the authors please briefly define Interpretive Phenomenological Analysis for the reader? One extra sentence will suffice. Also, in the Supplementary file 5 (Phase 2 questionnaire), can the authors please define Community of Practice for the questionnaire respondent? Finally, there are empty rows in the chart on p40 lines 6&7 which I believe should be removed. Great work! I look forward to reading the results at the end of the study.
--

VERSION 2 – AUTHOR RESPONSE

Reviewer 1

Comment	Response
Could the authors please briefly define Interpretive Phenomenological Analysis for the reader? One extra sentence will suffice.	The following text has been added in section 6.7.2 to define IPA: a qualitative research approach which is used to explore and examine personal lived experiences
Also, in the Supplementary file 5 (Phase 2 questionnaire), can the authors please define Community of Practice for the questionnaire respondent? Finally, there are empty rows in the chart on p40 lines 6&7 which I believe should be removed.	While S5 has been removed, the author takes note of this suggestion and will include a definition of Community of Practice at the start of the questionnaire for the benefit of the respondents. Thank you for this suggestion.